# Anderson–Fabry Disease: An Overview of Current Diagnosis, Arrhythmic Risk Stratification, and Therapeutic Strategies

**DOI:** 10.3390/diagnostics15020139

**Published:** 2025-01-09

**Authors:** Chiara Tognola, Giacomo Ruzzenenti, Alessandro Maloberti, Marisa Varrenti, Patrizio Mazzone, Cristina Giannattasio, Fabrizio Guarracini

**Affiliations:** 1Clinical Cardiology Unit, De Gasperis Cardio Center, Niguarda Hospital, 20162 Milan, Italy; 2School of Medicine and Surgery, University of Milano-Bicocca, 20126 Milan, Italy; 3Electrophysiology Unit, De Gasperis Cardio Center, Niguarda Hospital, 20162 Milan, Italy

**Keywords:** Anderson–Fabry disease, arrhythmic risk stratification, sudden cardiac death, conduction system disorders, tachyarrhythmias, atrial fibrillation

## Abstract

Anderson–Fabry disease (AFD) is a rare X-linked lysosomal storage disorder characterized by the accumulation of globotriaosylceramide, leading to multi-organ involvement and significant morbidity. Cardiovascular manifestations, particularly arrhythmias, are common and pose a considerable risk to affected individuals. This overview examines current approaches to arrhythmic risk stratification in AFD, focusing on the identification, assessment, and management of cardiac arrhythmias associated with the disease. We explore advancements in diagnostic techniques, including echocardiography, cardiac MRI, and ambulatory ECG monitoring, to enhance the detection of arrhythmogenic substrate. Furthermore, we discuss the role of genetic and biochemical markers in predicting arrhythmic risk and the implications for personalized treatment strategies. Current therapeutic interventions, including enzyme replacement therapy and antiarrhythmic medications, are reviewed in the context of their efficacy and limitations. Finally, we highlight ongoing research and future directions with the aim of improving arrhythmic risk assessment and management in AFD. This overview underscores the need for a multidisciplinary approach to optimize care and outcomes for patients with AFD.

## 1. Introduction

Anderson–Fabry disease (AFD) is a rare X-linked lysosomal storage disorder caused by variants in the GLA gene, leading to a deficiency of the enzyme α-galactosidase A. This enzymatic deficit results in the accumulation of globotriaosylceramide (Gb3) in various tissues, including the heart, kidneys, and nervous system. The incidence of AFD ranges from approximately 1 in 40,000 to 1 in 170,000 live births, with significant variability depending on the population studied. Newborn screening programs have shown that the prevalence of potentially pathogenic GLA variants is higher than previously thought, particularly in late-onset variants. Cardiac involvement is a major cause of morbidity and mortality in AFD. Cardiovascular manifestations typically include left-ventricular hypertrophy (LVH), arrhythmias, heart failure, and valvular disease. In the early stages, patients often present with LVH and preserved systolic function, mimicking hypertrophic cardiomyopathy. However, over time, the disease can lead to progressive myocardial dysfunction, small-vessel ischemia, and fibrosis. Arrhythmias, including atrial fibrillation and ventricular arrhythmias, are common in later stages, contributing to a higher risk of sudden cardiac death [1].

Advanced cardiac imaging techniques, such as speckle-tracking echocardiography and cardiac magnetic resonance imaging (MRI), have improved the ability to detect subclinical myocardial dysfunction in AFD patients. These methods are useful for assessing myocardial deformation and fibrosis, which are key indicators of disease progression [2].

Arrhythmic risk stratification in Anderson–Fabry disease (AFD) focuses on identifying patients at heightened risk of life-threatening arrhythmias, including ventricular tachycardia (VT) and sudden cardiac death (SCD). The development of arrhythmias is a significant concern due to progressive cardiac involvement typical of AFD, particularly left-ventricular hypertrophy (LVH), myocardial fibrosis, and conduction system abnormalities.

To stratify arrhythmic risk, several clinical and imaging parameters are used: (I) Electrocardiography (ECG) is used as AFD patients often exhibit abnormalities such as short PR intervals, repolarization abnormalities, and premature ventricular contractions (PVCs), all of which are markers of potential arrhythmic risk. (II) Cardiac magnetic resonance imaging (CMR) is pivotal in detecting myocardial fibrosis using late gadolinium enhancement (LGE), which is highly associated with arrhythmic events. Fibrosis in the basal inferolateral wall is particularly linked to VT and SCD. (III) Holter monitoring can reveal arrhythmias that are not always evident on a resting ECG, such as non-sustained ventricular tachycardia (NSVT), which can be predictive of more severe arrhythmic events. (IV) Echocardiography—beyond LVH, advanced techniques such as speckle-tracking echocardiography help identify subclinical ventricular dysfunction, which may also relate to arrhythmic risk [3].

Enzyme replacement therapy (ERT) is the cornerstone of AFD treatment, designed to replace the deficient α-galactosidase A enzyme and consequently slow down disease progression. By clearing accumulated Gb3 deposits, particularly in cardiac tissues, ERT can stabilize or even reverse left-ventricular hypertrophy and reduce myocardial fibrosis, which are critical factors in arrhythmia development [4].

## 2. Diagnosis

Cardiac involvement in AFD is common and progresses over time, becoming one of the primary causes of morbidity and mortality. A multi-modality approach, combining electrocardiography, echocardiography, cardiac magnetic resonance imaging, blood tests, and genetic analysis, is essential for early detection and accurate risk stratification. This chapter focuses on these key diagnostic tools and their significance in the evaluation of cardiac involvement in AFD.

### 2.1. Electrocardiography

ECG is a fundamental tool in the early detection of cardiac abnormalities in patients with AFD. The first sign of cardiac involvement often appears as a shortened PR interval, since glycosphingolipid accumulation speeds up conduction through the atrioventricular node. This feature is particularly important in younger patients, where cardiac symptoms may still be subtle. With disease progression, patients frequently show signs of left-ventricular hypertrophy (LVH) on ECG, such as increased QRS voltage and ST-segment depression or T-wave inversions, mainly in the precordial leads. Patients in advanced stages may develop bundle branch blocks, often right bundle branch block (RBBB), due to conduction disturbances. The progressive deposition of glycolipids in cardiac tissues leads to fibrosis and disruption of the normal conduction system, resulting in atrioventricular (AV) conduction delays and sometimes complete AV block. In some cases, atrial fibrillation (AF) develops, adding to the risk of stroke and further complicating the clinical management of these patients. ECG changes in AFD correlate closely with the extent of myocardial fibrosis, making it a useful tool for disease monitoring [5]. Table 1 shows ECG findings in patients affected by AFD.

### 2.2. Echocardiography

Echocardiography is an essential imaging modality in the diagnosis and monitoring of cardiac involvement in AFD. LVH is the hallmark finding and can be detected as early as the third decade in males and the fourth decade in females. The hypertrophy observed in AFD is usually concentric, affecting the entire ventricular wall, and is often accompanied by prominent papillary muscles, which may mimic hypertrophic cardiomyopathy (HCM). However, LVH in AFD tends to be more uniform, without the dynamic outflow obstruction seen in sarcomeric HCM. Advanced echocardiographic techniques, such as tissue Doppler imaging (TDI) and strain imaging, have been pivotal in detecting early signs of diastolic dysfunction, which is one of the earliest functional impairments in AFD-related cardiomyopathy. Diastolic dysfunction results from the stiffening of the left ventricle due to progressive glycolipid deposition and fibrosis. The combination of LVH and diastolic dysfunction is often accompanied by mitral valve abnormalities, such as mitral regurgitation, which can be detected through Doppler studies. These changes reflect the involvement of the mitral valve apparatus, especially in cases of extensive posterolateral fibrosis. In later stages, echocardiography can reveal regional wall motion abnormalities due to fibrosis, especially in the posterolateral basal segments. In severe cases, left-ventricular thinning and the development of left-ventricular aneurysms are seen. Additionally, right-ventricular hypertrophy (RVH) is frequently observed, although it rarely leads to significant clinical symptoms. Given the limitations of conventional echocardiography in detecting early fibrosis, speckle-tracking echocardiography and global longitudinal strain (GLS) measurements have gained prominence as sensitive tools to detect subclinical dysfunction before overt hypertrophy or fibrosis becomes evident [6]. Table 2 shows the main echo findings.

### 2.3. Cardiac Magnetic Resonance

CMR has emerged as a pivotal tool in the early differential diagnosis and staging of cardiac involvement in AFD. Characteristic findings include LGE, which typically appears in the mid-myocardial basal inferolateral wall and reduced native T1 values. These changes are believed to reflect the accumulation of glycosphingolipids within myocardial tissue and can precede the development of significant LVH [7]. Unlike cardiac amyloidosis, which primarily involves extracellular deposits, AFD is predominantly an intracellular storage disorder. As a result, the extracellular volume (ECV) remains normal in most areas, except in regions showing LGE. The typical localization of LGE in the basal posterolateral left ventricle is characteristic of AFD and helps differentiate it from other causes of LVH, such as hypertrophic cardiomyopathy. T1 mapping has proven useful for detecting intracellular glycosphingolipid storage in the myocardium even before LGE becomes apparent, making it a valuable tool for early diagnosis [7]. In advanced stages, it is possible to observe a pseudo-normalization of T1 values and LGE with wall thinning in the basal inferolateral segment. Progressive T1 pseudo-normalization in advanced phases may be caused by increased interstitial and replacement fibrosis, myocardial inflammation, and myocardial hypertrophy relative to the storage component [7].

The degree of myocardial fibrosis detected by LGE and T1 mapping correlates with disease severity and prognosis. Patients with extensive LGE are at higher risk for ventricular arrhythmias and sudden cardiac death (SCD) [8].

A recent study [9] showed that segments with LGE exhibited significantly elevated T2 values. These elevations were associated with increased levels of troponin and N-terminal pro–B-type natriuretic peptide, electrocardiographic abnormalities, and reduced global longitudinal strain. In these patients, both localized (LGE-related) and global T2 elevations were higher than those observed in other myocardial conditions, such as sarcomeric HCM. Persistent T2 elevation and elevated troponin levels over a one-year period suggested ongoing myocardial edema and damage, which were linked to clinical deterioration. If validated by histological analysis or other methods, these findings could underscore inflammation as a key contributor to AFD pathogenesis, potentially paving the way for targeted therapeutic strategies.

Moreover, CMR is invaluable for accurately assessing left-ventricular mass and evaluating ventricular function. Patients with AFD often demonstrate preserved ejection fraction in the early stages, but as fibrosis progresses, systolic dysfunction and reduced global longitudinal strain become apparent.

CMR is also effective in evaluating right-ventricular involvement, which is common but often underdiagnosed due to the limitations of echocardiography.

The advanced imaging resolution of CMR allows for a more thorough evaluation of both left- and right-ventricular function, offering a more comprehensive view of the heart’s structural and functional changes.

The application of artificial intelligence (AI) to CMR is an emerging field that is being developed to enhance the diagnosis and prognosis of various cardiac conditions. In the context of Fabry disease, AI-based CMR and transcriptome correlation analyses have shown potential in identifying biomarkers for cardiac complications. For example, the CHN1 gene, which encodes chimerin 1 involved in myocardial fibrosis, was identified as a potential predictor of cardiac complications in Fabry disease through AI-driven MRI analysis and RNA transcriptome data extracted from the peripheral blood of Fabry disease patients and healthy individuals [10]. Table 3 shows the main CMR findings and advantages of CMR compared to echo.

### 2.4. Blood Tests

Biochemical analysis plays a crucial role in diagnosing AFD, especially in screening for the disease. The most widely used test is the measurement of α-galactosidase A enzyme activity in plasma or leukocytes, which is markedly reduced in males with AFD. However, female carriers often have normal or borderline enzyme levels, necessitating further diagnostic testing. Additionally, plasma lyso-Gb3 levels, a biomarker of disease activity, are elevated in both male and female individuals and correlate with the severity of cardiac and extracardiac manifestations.

### 2.5. Genetic Testing

A definitive diagnosis of AFD is made through the genetic testing of the GLA gene, which encodes the α-galactosidase A enzyme. The diagnosis of AFD varies on the patient’s gender: in male patients, reduced or absent α-Gal A activity is diagnostic and the identification of a hemizygous GLA pathogenic variant by molecular genetic testing confirms the diagnosis. In female patients, the identification of a heterozygous GLA pathogenic variant by molecular genetic testing establishes the diagnosis; in fact, woman can carry heterozygous mutations in the GLA gene, and because of skewed X-chromosome inactivation, they may have normal enzyme activity, making genetic testing crucial [11].

The GLA gene encodes the lysosomal enzyme α-Gal A and spans 10,223 base pairs, consisting of seven exons and six introns, which together produce a 1290-nucleotide transcript. Variations in the GLA gene can lead to reduced or absent α-Gal A activity, impairing the cell’s ability to break down glycosphingolipids that contain terminal d-galactose residues, such as globotriaosylceramide and its derivatives. As a result, these substrates accumulate in the cytoplasm and lysosomes of various tissues, contributing to the clinical manifestations of the disease. Enzyme activity can be impaired in several ways: (1) modifications to essential residues at the enzyme’s active site; (2) variants affecting buried residues, distant from the active site, which can disrupt protein folding and stability; (3) the disruption of critical disulfide bonds or loss of N-glycosylation sites can prevent the enzyme from entering the lysosome, resulting in a loss of enzyme activity. However, a recent article identified a deep intronic variant (c.640–814T>C) located at intron 4, which is associated with α-GalA deficiency, predominantly manifesting as the cardiac form of FD [12]. This variant disrupts splicing, resulting in a truncated, non-functional protein that may have a dominant-negative effect on the wild-type GLA enzyme. These findings underscore the significance of detecting deep intronic variants in the diagnosis of Fabry disease, as they might be missed by whole-exome sequencing.

Genetic testing is especially important for female carriers and individuals with borderline enzyme activity, as it helps identify those at risk for developing significant cardiac involvement. Furthermore, identifying the specific GLA variant may provide prognostic information, as certain variants are associated with more severe cardiac phenotypes. For instance, the c.337T>C (p.F113L) variant is linked to a late-onset phenotype with predominant cardiac manifestations [13]. Additionally, the T410A variant has been observed to cause late-onset AFD with progressive cardiomyopathy in older males and classic AFD in younger males within the same family [14].

### 2.6. Extracardiac Manifestations

Extracardiac manifestations are crucial in diagnosing AFD, as they often precede cardiac involvement. They are diverse and can significantly impact patients’ quality of life. Common features include neurological symptoms, renal dysfunction, dermatological, gastrointestinal, and ocular manifestations, and hearing loss. The early recognition and management of these manifestations can prompt early diagnosis and treatment.

Regular monitoring and a multidisciplinary approach involving nephrologists, neurologists, dermatologists, and genetic counselors are needed for comprehensive patient care.

Neurological symptoms in AFD often include pain episodes: patients frequently experience recurrent episodes of acroparesthesia, often described as burning or tingling sensations in the extremities. This symptom is a result of small nerve fiber damage due to Gb3 accumulation. Moreover, AFD patients are at increased risk for cerebrovascular events such as transient ischemic attacks (TIAs) and strokes, primarily due to the endothelial dysfunction and atherosclerosis associated with the disease. Finally, dysautonomia due to autonomic neuropathy can lead to symptoms such as gastrointestinal dysmotility, urinary incontinence, and orthostatic hypotension.

Progressive renal impairment is common, often leading to end-stage renal disease (ESRD). Gb3 accumulation can damage renal tubules and glomeruli, impairing kidney function. Patients may exhibit proteinuria, often a sign of glomerular damage. Renal impairment, added to sympathetic nervous system involvement, can contribute to hypertension.

Dermatological manifestations of AFD are particularly distinctive and may include angiokeratomas, small, dark-red-to-purple lesions that are typically located in the bathing trunk area and represent a classic dermatological sign of AFD. Patients often experience reduced sweating (hypohidrosis), particularly in the lower extremities, which can lead to heat intolerance.

Gastrointestinal symptoms may include diarrhea, constipation, and abdominal pain due to autonomic dysfunction.

Ocular involvement in AFD is less common but can be significant, leading to corneal deposits and retinal changes associated with vascular occlusions.

In the end, sensorineural hearing loss can occur in AFD due to neuronal damage and has been documented in both male and female individuals with the disease.

## 3. Arrhythmic Risk Stratification

In AFD, beyond the better-known complication of heart failure, there is increasing evidence that arrhythmias and SCD may represent frequent clinical presentations [15]. Depending on gender and disease stage, 15% to 43% of adult patients report having palpitations [9], with atrial arrhythmia being described as the most frequent cause [16]. Between 3.6% and 5.6% of male patients and 1.7% and 2.6% of female patients have a history of syncope. Syncope may be caused by autonomic dysfunction, sinus node dysfunction, complete heart block, or sustained ventricular tachyarrhythmia [17]. Death in AFD is mainly CV (75%), with SCD being the most common cause of death (62%) [3]; from a mortality perspective, AFD may be considered a predominantly cardiac disease.

Tachyarrhythmias are common and may be supraventricular or ventricular in origin. With AFD being a progressive storage disease, arrhythmias are more common in older patients with more extensive fibrosis [18,19]. Atrial fibrillation is the most frequently reported supraventricular arrhythmia, with a prevalence of 5% of males and 3% of females and an incidence of 6% per annum [20,21]; left-ventricular hypertrophy with diastolic dysfunction, LA dilatation, and impaired LA strain favor atrial fibrillation. However, direct atrial sphingolipid accumulation may contribute to LA dysfunction and predispose an individual to atrial fibrillation [8]. Ventricular tachycardia and ventricular fibrillation have an overall prevalence between 13 and 18% [21]. SCD incidence varies between 0.5 and 3% (follow-up range, 1.2–8 years) [20]. AFD patients, in particular male patients with classical AFD, seem to have a much higher risk of VA and SCD than sarcomeric hypertrophic cardiomyopathy [22].

Bradyarrhythmias, such as sinus node and atrioventricular node disease, are more frequent than tachyarrhythmias [23]. The detection rate of arrhythmias varies with the length of monitoring; current guidelines advise annual Holter monitoring, which is likely to underestimate the frequency of arrhythmias [24]. Symptomatic bradycardia caused by sinus node dysfunction and AV block is relatively common in AFD with the need for pacing best predicted by QRS duration and PR interval [25]. Bradycardia should be treated in accordance with the current ESC guidelines (2021). If AV block is caused by drugs, then their indication should be reviewed and the need for pacing re-evaluated thereafter.

The pathogenesis of arrhythmias comprises several hypotheses. Glycosphingolipid accumulation alters the expression of calcium and sodium ion channels, altering cellular electrophysiological properties [26,27]. In a cellular model, Gb3 accumulation led to a more positive diastolic membrane potential, faster action potential upstroke velocity, a greater burden of delayed afterdepolarizations, a greater contraction force, a slower beat rate, and dysfunction in calcium handling [28,29]. Cardiac sphingolipid accumulation triggers myocyte hypertrophy and the development of interstitial myocardial fibrosis; in fact, sphingolipid accounts for only 5% of the increase in ventricular mass [30]. Myocardial structure disarray predisposes individuals to arrhythmias. Additionally, glycosphingolipid lysosomal storage causes cell apoptosis and pro-inflammatory cytokine secretion with myocardial edema. Elevated T2 on magnetic resonance may be present in the inferolateral wall and correlate closely to increased high-sensitivity troponin and pro-inflammatory cytokines such as interleukin 6, even before the appearance of LVH and LGE [9]. In this setting, myocyte turnover was associated with plasma hs-troponin T elevation, the values of which predicted the extent of fibrosis [31]. Associated renal dysfunction may trigger arrhythmias including AF, bradyarrhythmia, VA, and SCD due to the effects of electrolyte imbalance, as well as a rapid fluctuation in electrolyte levels and hemodynamic shifts during dialysis [32].

AFD therapy should be started as soon as possible because, once cardiomyopathy has developed and arrhythmic risk is increased, the benefits of ERT and OCT initiation are significantly limited. However, there is currently no proof that ERT reduces the arrhythmic burden in AFD [33,34].

An arrhythmic presentation may be even used to select a population to screen for AFD. Arrhythmias may occur before other evidence of AFD cardiomyopathy, such as severe left-ventricular hypertrophy, because sphingolipid deposition in conduction tissue occurs at an early stage of disease [35]. Hemelsoet et al. [36] tested a total of 531 male patients implanted with a pacemaker (68%) or implantable cardioverter–defibrillator (32%) between 30 and 76 years old (mean, 59.8 years) to screen for AFD with the determination of α-Gal A enzyme activity; patients with low or absent enzymatic activity were subjected to DNA analysis. In three unrelated patients (0.56% of the total cohort), a GLA variant was identified, confirming a diagnosis of Fabry disease.

Supraventricular tachycardia and AF may be poorly tolerated by AFD patients due to diastolic dysfunction; rhythm control is therefore the optimal strategy. The maintenance of sinus rhythm is often challenging due to an evolving atrial substrate and the significant limitations of the available drugs. There is general support for early ablation in all patients presenting with AF, but concerns remain regarding long-term efficacy in AFD; extrapolating data from HCM, a high rate of AF relapse with a need for repeated procedures is expected, particularly in older people with left-atrial dilatation. Among drugs for rhythm control, long-term amiodarone is generally avoided as it may alter lysosomal pH and enzyme activity and may induce phospholipidosis via the inhibition of lysosomal phospholipase activity [37]. When a rate control strategy is chosen, caution is needed because of the risk of bradyarrhythmia and conduction disease. Anticoagulation is always recommended in AFD patients with a history of AF. Since AFD may be associated with cerebral micro-bleeding, direct oral anticoagulation is preferred due to lower risk of intracranial bleeding.

The risk stratification of ventricular arrhythmias is difficult due to the paucity of data. Left-ventricular hypertrophy, cardiac fibrosis, and NSVT are the main risk factors for arrhythmic cardiac death. The absence of left-ventricular hypertrophy and late gadolinium enhancement seem to correlate with a low arrhythmic risk [38]. When considering implanting a device, we should remember that the substrate changes in AFD place the patient at increased risk of any arrythmia; this suggests that caution should be exercised in implanting cardiac devices without a defibrillator function even in the absence of VA. Once a substrate has been created, no target treatment has proved to be effective in reducing it, so early diagnosis is essential in preventing further damage.

An ICD is recommended in patients with a life expectancy >1 year who have survived a cardiac arrest due to ventricular tachycardia or fibrillation or who have spontaneous sustained ventricular tachycardia causing hemodynamic compromise. In a primary prevention setting, in contrast to hypertrophic cardiomyopathy, in AFD, there is no validated prediction model to evaluate risk of SCD. Advanced LVH and extensive–rapidly progressing fibrosis may be considered for ICD implantation. Patients with significant LVH and unexplained syncope may also be considered for ICD implantation. However, most implantable cardioverter defibrillators placed in patients with AFD are in secondary prevention [39]. Existing approaches to SCD prevention face major challenges, such as the absence of comprehensive data from large-scale prospective studies, the lack of a validated SCD prediction model for risk assessment, and the use of composite or surrogate endpoints in current studies [40]. Asymptomatic NSVT does not usually require anti-arrhythmic therapy, and its relationship with sudden cardiac death (SCD) risk is unknown, but, as in most myocardial diseases, it may be related to the fibrosis extent and therefore to SCD. The catheter ablation of monomorphic VT has been described for patients with recurrent ventricular tachycardia, using a combined endocardial–epicardial approach since the site of origin is frequently deep within the myocardium [30].

Hanneman et al. [41], in their study with 90 AFD patients, identified LGE (HR, 7.2; 95% CI, 1.5, 34; *p* = 0.01) and LVH (HR, 3.0; 95% CI, 1.1, 8.1; *p* = 0.03;) as predictors of a combined end-point of NSVT or SVT, bradycardia requiring device implantation for pacing, severe heart failure with NYHA III or IV symptoms, and cardiac death. The risk also increased with the LGE extent: for every 5% increase in LGE extent, there was a 44% higher risk of adverse cardiac events, and patients with an LGE extent of 15% or greater had a 12 times higher risk of adverse cardiac events compared with those without LGE. The absence of both LVH and LGE on cardiac MRI is associated with low risk. LGE may not be modified by disease-specific therapy. The incidence composite end-point was 7.6% per year, and non-sustained ventricular tachycardia mostly contributed to the composite end-point.

Baig et al. [17] identified male gender, older age (>40 years in male individuals), an increasing LV mass index, the presence of LGE on MRI, and prior NSVT as predictors of SCD. AFD is more age-dependent than HCM, and advancing age predicts an increased risk of SCD. This is consistent with the concept of AFD as a progressive myocardial storage disease. This relationship is less clear in women and can be seen 10–15 years later, in comparison to >40 years old in men. NSVT has been associated with an increased risk of SCD but its usefulness in selecting patients for ICD is less well defined and needs to be addressed in further studies. Like HCM, the risk of SCD is linked to LVH, but, as cited earlier, hypertrophy in AFD is not simply a proportionate consequence of sphingolipid deposition. In AFD, LGE is intramural or sub-epicardial with the predominant involvement of basal inferolateral LV, and not only end-stage fibrosis but also acute inflammation may be present, with high T2 values reflecting edema.

Orsborne et al. [42], in a longitudinal prospective cohort study with 200 consecutive AFD patients, developed a risk prediction model that accurately predicted the 5-year risk of an adverse cardiac outcome (first hospitalization for heart failure, myocardial infarction, coronary revascularization, ventricular tachycardia, new atrial fibrillation, bradyarrhythmia necessitating pacemaker implantation, aborted sudden cardiac death, appropriate ICD therapy, or cardiovascular death). The risk model was simple to integrate into clinical care and was internally validated; external validation is warranted. The adjusted c-statistic was 0.77 (95% CI: 0.70–0.84). The highest-performing internally validated parsimonious multivariable model included age, native myocardial T1 dispersion, and indexed left-ventricular mass. This is the first study to examine the prognostic value of myocardial T1 measurements in AFD. The shortening of T1 is thought to reflect the accumulation of glycosphingolipid. However, myocardial fibrosis and inflammation are both associated with increased native T1, potentially reducing the prognostic impact of T1. Therefore, Osborne et al. considered T1 dispersion, which may better reflect disease severity. Composite outcome was mainly due to NSVT. Given there was no sudden cardiac death or sustained ventricular arrhythmias, the risk model estimator could not be used to guide device therapy. Table 4 shows studies published in the literature on arrhythmic risk stratification in AFD. Table 5 shows the main parameters for defining arrhythmic risk in AFD.

## 4. Therapeutic Approaches

### 4.1. Enzyme Replacement Therapy (ERT)

The advent of enzyme replacement therapy has significantly altered the treatment landscape for AFD. Two recombinant enzyme therapies are currently available: agalsidase alfa and agalsidase beta. They aim to restore deficient α-galactosidase A activity, which can help reduce the accumulation of Gb3 in tissues. The benefits of ERT in cardiac manifestations of AFD have been well documented. ERT helps to stabilize or even regress LVH, which is a hallmark of AFD-related cardiomyopathy, and may reduce the extent of myocardial fibrosis, a significant contributor to arrhythmic risk [43,44]. In several longitudinal studies, patients treated with ERT demonstrated improvements in cardiac morphology and function, particularly regarding left-ventricular mass (LVM) and myocardial strain. For instance in a recent study patients on long-term ERT showed a reduction in left-ventricular mass index (LVMI), particularly if therapy was initiated early, before the onset of extensive fibrosis. The earlier ERT is initiated, the more effective it is in mitigating progression towards irreversible fibrosis [45]. Hopkin et al. analyzed data from patients enrolled in the Fabry Registry [46] with classic or unclassified GLA variants, who began agalsidase beta treatment before the age of 30. Echocardiographic measures, such as the end-diastolic interventricular septal thickness and left-ventricular posterior wall thickness (median follow-up of ≥4 years), remained stable in both male individuals and younger female individuals. Additionally, reports of Fabry disease symptoms decreased significantly, especially among males during long-term follow-up [47]. Recent data from the Fabry Outcome Survey (FOS) involving 1374 patients (172 who were promptly treated and 1202 who experienced delays; 807 males, 567 females) based on the time from symptom onset, and 2051 patients (1106 promptly treated, 1045 delayed; 1130 male, 921 female) based on the time from diagnosis, revealed significant benefits associated with early ERT initiation. In the overall cohort, prompt treatment was linked to a notable reduction in cardiovascular events—both from symptom onset (hazard ratio [HR] = 0.62; *p* < 0.001) and from diagnosis (HR = 0.83; *p* < 0.003). These benefits were observed in both male and female patients with Fabry disease (female vs. male patients: symptom HR = 0.83, *p* = 0.018; diagnosis HR = 0.82, *p* = 0.003) [48]. Additionally, an analysis based on age at symptom onset indicated that the positive effects of early ERT initiation were more pronounced in patients who were ≤20 years old at symptom onset. Moreover, reductions in myocardial Gb3 deposits have been associated with improvements in electrophysiological stability, as documented by Holter monitoring and reductions in premature ventricular contractions (PVCs), which are often an early sign of ventricular arrhythmias in AFD patients. A recent study demonstrated that patients treated with ERT had reduced LVH and a lower incidence of NSVT, a predictor of sudden cardiac death in AFD patients. Furthermore, long-term ERT has been shown to reduce the burden of atrial arrhythmias, particularly in the early stages of the disease [45]. However, the impact of ERT on advanced cardiac disease and arrhythmias is limited. In patients with significant myocardial fibrosis, ERT alone may not reverse the arrhythmic substrate. For these individuals, adjunctive treatments, including antiarrhythmic drugs and implantable cardioverter–defibrillators (ICDs), are often necessary. ERT’s efficacy in reducing arrhythmic events is highly dependent on the timing of therapy initiation [49]. Early treatment, particularly in the preclinical or early hypertrophic stages, has been shown to stabilize or even regress hypertrophy and reduce the development of fibrosis [50,51,52]. However, in more advanced stages, particularly in patients with significant myocardial fibrosis, the benefit is limited [53]. This is partly due to the inability of ERT to adequately penetrate fibrotic tissue.

The recommended follow-up should be every 6 months in patients receiving ERT. The aims of follow-up are to assess treatment efficacy and safety, and it provides an opportunity to confirm that the patient is taking optimal adjunctive therapy in terms of renoprotective, cardioprotective, and vasculoprotective medication. According to our review findings, we believe that CMR should be systematically, and maybe more frequently, integrated into monitoring Fabry patients to improve risk stratification and optimize therapeutic management, particularly in those with signs of advanced cardiac disease or high risk of arrhythmic complications. Table 6 summarizes the recommended cardiac follow-up for patients on ERT.

### 4.2. Chaperone Therapy: Migalastat

Another therapeutic option that has gained attention in the treatment of AFD is migalastat, a pharmacological chaperone designed to stabilize certain mutant forms of α-galactosidase A, enhancing their residual enzymatic activity. Unlike ERT, which involves exogenous enzyme administration, migalastat works by binding to the patient’s own enzyme, thereby improving its stability and trafficking to the lysosome. It stabilizes the misfolded alpha-galactosidase A enzyme, facilitating its proper trafficking to the lysosome where it can degrade Gb3. Studies on the cardiac effects of migalastat are promising, particularly in patients with amenable variants, meaning variants that respond to migalastat therapy by restoring enzymatic activity. In the ATTRACT trial, patients treated with migalastat showed a stabilization of LVH and, in some cases, even the regression of left-ventricular mass [54]. These findings were corroborated by reductions in plasma lyso-Gb3 levels, suggesting the effective clearance of Gb3 from cardiac tissues. This trial was pivotal in demonstrating the efficacy of migalastat in reducing left-ventricular mass and stabilizing kidney function in patients with amenable variants. Importantly, the trial highlighted the fact that migalastat was associated with a stabilization of cardiac arrhythmias, including AF and ventricular ectopy, when compared to a placebo. In terms of arrhythmic risk, the reduction in LVH and myocardial strain associated with migalastat may contribute to improved electrophysiological stability, although the long-term data on its impact on ventricular arrhythmias and SCD are still emerging. Preliminary studies suggest that patients on migalastat have a lower incidence of arrhythmias than untreated patients, likely due to the reduction in substrate accumulation and fibrosis [44,45]. However, it remains to be seen whether migalastat can prevent the development of myocardial fibrosis, a key substrate for arrhythmias in AFD [12]. Migalastat offers an alternative to ERT for patients with specific variants and has shown promise in reducing the burden of arrhythmias, particularly in early-stage disease. The drug’s oral administration and ability to reach cardiac tissues make it a convenient and potentially more effective option for long-term management. While migalastat has demonstrated efficacy in patients with amenable variants, its use is restricted to this subgroup. Moreover, studies are ongoing to evaluate its long-term impact in preventing serious arrhythmic events such as SCD.

### 4.3. Supportive Treatment for Arrhythmia Management

In addition to disease-specific therapies like ERT and migalastat, supportive treatments play a crucial role in managing arrhythmias and other cardiac complications in AFD. These treatments include the use of antiarrhythmic drugs, devices (such as pacemakers and implantable cardioverter–defibrillators), and catheter ablation for managing persistent or life-threatening arrhythmias. Antiarrhythmic drugs such as amiodarone and beta-blockers are commonly used to manage atrial fibrillation and ventricular arrhythmias in AFD patients. However, their efficacy can be limited, particularly in the presence of significant myocardial fibrosis. Additionally, the use of amiodarone carries the risk of long-term side effects, and its use in younger patients with AFD requires careful consideration. Beta-blockers are often preferred as first-line agents for managing arrhythmias, particularly in patients with LVH and diastolic dysfunction, which are common in AFD cardiomyopathy.

For patients at high risk of SCD due to ventricular arrhythmias or those with advanced conduction system disease (e.g., complete AV block), the implantation of ICDs or pacemakers is a vital intervention. ICDs have been shown to reduce mortality in AFD patients with a history of VT or non-sustained ventricular tachycardia (NSVT) detected on Holter monitoring or ILRs [16]. In cases where fibrosis leads to significant conduction system disease, pacemakers may be necessary to maintain an adequate heart rate and prevent bradyarrhythmias [52]. The decision to implant an ICD or pacemaker is typically guided by risk stratification tools, including CMR findings, particularly the extent of LGE and T1 mapping abnormalities, as well as the presence of documented arrhythmias on Holter monitoring or ILR recordings. The presence of LGE, even in the absence of symptomatic arrhythmias, is a strong predictor of future ventricular arrhythmias and is often considered an indication for ICD implantation in AFD patients. For patients with recurrent or drug-refractory arrhythmias, catheter ablation offers another therapeutic option. While the success rates of ablation procedures in AFD patients can be lower than in patients without underlying cardiomyopathy, the ablation of AF and VT has been successfully performed in select cases. In a recent study, catheter ablation for AF in AFD patients resulted in a reduced arrhythmia burden and improved quality of life, although the recurrence rates were higher than in the general population due to the presence of underlying fibrosis [55].

### 4.4. Gene Therapy

Gene therapy is an emerging therapeutic strategy that holds the potential to provide a definitive cure for AFD. Preclinical studies have demonstrated the feasibility of using viral vectors to deliver a functional copy of the GLA gene to affected cells, resulting in sustained alpha-galactosidase A activity and a reduction in Gb3 accumulation.

While it is still in the experimental phase, gene therapy may offer a unique advantage in preventing arrhythmias by addressing the root cause of AFD at a genetic level. Animal models have shown that early intervention with gene therapy can prevent the development of LVH and myocardial fibrosis, key contributors to arrhythmic risk. If these findings could be translated to human trials, then gene therapy could represent a paradigm shift in the treatment of AFD, offering the potential to eliminate both the cardiac and extracardiac manifestations of the disease.

## 5. Conclusions

Arrhythmic risk stratification in AFD remains an evolving area, complicated by the interplay of progressive myocardial fibrosis, left-ventricular hypertrophy, and early conduction system abnormalities. While current imaging techniques, especially cardiac MRI with LGE and T1 mapping, provide valuable insights into myocardial fibrosis, these methods may not fully capture the underlying electrophysiological risk, particularly in early-stage disease.

The high incidence of both atrial and ventricular arrhythmias underscores the need for a more robust, electrophysiologically informed stratification model. Traditional risk factors, such as the extent of fibrosis and ventricular hypertrophy, are helpful but insufficient for accurately predicting arrhythmic events like SCD and sustained VT. EPS and advanced arrhythmic monitoring, including intracardiac mapping, could offer greater precision in identifying high-risk individuals and provide a deeper understanding of arrhythmogenic mechanisms in AFD.

Future studies should aim to refine stratification protocols by integrating electrophysiological data into imaging biomarkers, potentially guiding decisions for device implantation, ablation therapies, and more targeted arrhythmia management strategies. Improved risk stratification will be critical in preventing severe arrhythmias and optimizing long-term outcomes for AFD patients, highlighting the need for interdisciplinary research and collaboration between clinical imaging cardiologists, electrophysiologists, and genetic specialists.

## Figures and Tables

**Table 1 diagnostics-15-00139-t001:** ECG findings.

ECG Findings
Shortened PR Interval
Signs of Left-Ventricular Hypertrophy
Bundle Branch Blocks
Atrioventricular Conduction Delay
Atrial Fibrillation

**Table 2 diagnostics-15-00139-t002:** Echo findings.

Echo Findings
Left-Ventricular Hypertrophy
Prominent Papillary Muscles
Diastolic Dysfunction
Mitral Valve Abnormalities
Regional Wall Motion Abnormalities
Left-Ventricular Thinning
Left-Ventricular Aneurysms
Right-Ventricular Hypertrophy
Speckle-Tracking Echocardiography

**Table 3 diagnostics-15-00139-t003:** CMR findings and advantages.

CMR Findings
Basal Posterolateral Late Gadolinium Enhancement
Mid-Myocardial Fibrosis Pattern
Low T1 mapping values
Left-Ventricular Mass Measurement
Ventricular Function Assessment
Right-Ventricular Assessment

**Table 4 diagnostics-15-00139-t004:** Studies on arrhythmic risk stratification.

Study	Design	Population	Outcome	Predictors and HR
Hanneman et al. [41]	Retrospective cohort	90	Composite outcome (VT, bradycardia requiring PM, severe HR, cardiac death)	LVH (3.0; CI, 1.1–8.1), LGE (7.2; CI, 1.5–34).
Baig et al. [17]	Systematic review	4185	SCD and VA	Male gender, older age (>40 years in males), increasing LV mass index, LGE on CMR, NSVT
Orsborne et al. [42]	Prospective cohort	200	5 y composite outcome (first HF hospitalization, AMI, coronary revascularization, NSVT, SVT, new AF, bradyarrhythmia necessitating PM, aborted SCD, appropriate ICD therapy, cardiovascular death)	Age (1.049; CI, 1.022–1.077), native myocardial T1 dispersion (1.012; CI, 1.002–1.021), indexed LV mass (1.008; CI, 1.003–1.014)

List of abbreviations: AF, atrial fibrillation; AMI, acute myocardial infarction; HF, heart failure; ICD, implantable cardioverter defibrillator; LGE, late gadolinium enhancement; LV, left ventricle; LVH, left-ventricular hypertrophy; NSVT, non-sustained ventricular tachycardia; PM, pacemaker; SCD, sudden cardiac death; SVT, sustained ventricular tachycardia; VA, ventricular arrhythmia; VT, ventricular tachycardia.

**Table 5 diagnostics-15-00139-t005:** Main parameters for defining arrhythmic risk in AFD.

Main Parameters for Defining Arrhythmic Risk
Previous cardiac arrest
Extensive–rapidly progressing fibrosis on CMR
Advanced LVH on CMR
NSVT
Unexplained syncope
Age > 40 years old
Male gender
Native myocardial T1 dispersion on CMR
LA dysfunction (AF)

List of abbreviations: AF, atrial fibrillation; AFD, Anderson–Fabry disease; LA, left atrium; LVH, left-ventricular hypertrophy; CMR, cardiac magnetic resonance; NSVT, non-sustained ventricular tachycardia.

**Table 6 diagnostics-15-00139-t006:** Recommended follow-up for patients on ERT.

Every 6 months: medical history and concomitant medications, clinical examination, vital parameters, blood and urine tests, ECG
Every 12 months: as for 6 months + echocardiogram, 24 h ECG, CMR (if abnormal at baseline; if normal at baseline, then every 2 years)

## Data Availability

Not applicable.

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
