# Peer review of "Anderson–Fabry Disease: An Overview of Current Diagnosis, Arrhythmic Risk Stratification, and Therapeutic Strategies"

_diagnostics, 2025, doi:10.3390/diagnostics15020139_

Round 1
Reviewer 1 Report
Comments and Suggestions for Authors
Dear Authors,
Thank you for your manuscript titled "Arrhythmic Risk Stratification of Anderson-Fabry Disease: A Current Overview."
My comments are as follows:
1. The manuscript provides a broad narrative overview of the subject but does not focus as closely on the topic of arrhythmic risk stratification as suggested by the title. To improve alignment, I would recommend either refining the content to address arrhythmic risk stratification more directly or adjusting the title to reflect the scope of the review better.
2. To enhance the clarity and utility of your review, I suggest including a table that outlines the main parameters for defining arrhythmic risk in Anderson-Fabry disease, ordered by their importance. It would be helpful to cite relevant references for each parameter in the table.
3. If you wish to continue providing a more general narrative review on cardiovascular involvement in Fabry disease, I would recommend incorporating the following recent studies to enhance the comprehensiveness of your review further:
· Pieroni M, Moon JC, Arbustini E, Barriales-Villa R, Camporeale A, Vujkovac AC, Elliott PM, Hagege A, Kuusisto J, Linhart A, Nordbeck P, Olivotto I, Pietilä-Effati P, Namdar M. Cardiac Involvement in Fabry Disease: JACC Review Topic of the Week. J Am Coll Cardiol. 2021 Feb 23;77(7):922-936. doi: 10.1016/j.jacc.2020.12.024. PMID: 33602475.
· Rubino M, Monda E, Lioncino M, Caiazza M, Palmiero G, Dongiglio F, Fusco A, Cirillo A, Cesaro A, Capodicasa L, Mazzella M, Chiosi F, Orabona P, Bossone E, Calabrò P, Pisani A, Germain DP, Biagini E, Pieroni M, Limongelli G. Diagnosis and Management of Cardiovascular Involvement in Fabry Disease. Heart Fail Clin. 2022 Jan;18(1):39-49. doi: 10.1016/j.hfc.2021.07.005. Epub 2021 Oct 25. PMID: 34776082.
· Piccolo S, Casal M, Rossi V, Ferrigni F, Piccoli A, Bolzan B, Setti M, Butturini C, Benfari G, Ferrero V, Franchi E, Tomasi L, Ribichini FL, Mugnai G. Ventricular arrhythmias and primary prevention of sudden cardiac death in Anderson-Fabry disease. Int J Cardiol. 2024 Nov 15;415:132444. doi: 10.1016/j.ijcard.2024.132444. Epub 2024 Aug 13. PMID: 39128566.
· Monda E, Limongelli G. Sudden cardiac death risk prediction in Fabry disease: How many strings do we have on our bow? Int J Cardiol. 2025 Jan 1;418:132592. doi: 10.1016/j.ijcard.2024.132592. Epub 2024 Sep 24.
· Kobayashi H, Nakata N, Izuka S, Hongo K, Nishikawa M. Using artificial intelligence and promoter-level transcriptome analysis to identify a biomarker as a possible prognostic predictor of cardiac complications in male patients with Fabry disease. Mol Genet Metab Rep. 2024 Oct 13;41:101152. doi: 10.1016/j.ymgmr.2024.101152. PMID: 39484074; PMCID: PMC11525769.
· Beck M, Ramaswami U, Hernberg-Ståhl E, Hughes DA, Kampmann C, Mehta AB, Nicholls K, Niu DM, Pintos-Morell G, Reisin R, West ML, Schenk J, Anagnostopoulou C, Botha J, Giugliani R. Twenty Years of the Fabry Outcome Survey (FOS): Insights, Achievements, and Lessons Learned from a Global Patient Registry. Orphanet J Rare Dis. 2022 Jun 20;17(1):238. doi: 10.1186/s13023-022-02392-9. PMID: 35725623; PMCID: PMC9208147.
· Wanner C, Ortiz A, Wilcox WR, Hopkin RJ, Johnson J, Ponce E, Ebels JT, Batista JL, Maski M, Politei JM, Martins AM, Banikazemi M, Linhart A, Mauer M, Oliveira JP, Weidemann F, Germain DP. Global reach of over 20 years of experience in the patient-centered Fabry Registry: Advancement of Fabry disease expertise and dissemination of real-world evidence to the Fabry community. Mol Genet Metab. 2023 Jul;139(3):107603. doi: 10.1016/j.ymgme.2023.107603. Epub 2023 Apr 29. PMID: 37236007.
· Zhang P, Wang Y, Jiang G, Zhang Y, Chen Y, Peng Y, Chen Z, Bai M. The c.640-814T>C mutation in the deep intronic region of the alpha-galactosidase A gene is associated with Fabry disease via a dominant-negative effect. Gene. 2025 Feb 5;936:149127. doi: 10.1016/j.gene.2024.149127. Epub 2024 Nov 28. PMID: 39613053.
Thank you.
Kind regards,
Reviewer
Author Response
Thank you very much for taking the time to review this manuscript. Please find the detailed responses below and the corresponding revisions in track changes in the re-submitted file.
Comments 1: The manuscript provides a broad narrative overview of the subject but does not focus as closely on the topic of arrhythmic risk stratification as suggested by the title. To improve alignment, I would recommend either refining the content to address arrhythmic risk stratification more directly or adjusting the title to reflect the scope of the review better.
Response 1: We agree with this comment. Therefore, we have s modified the title to better reflect the scope of our review.
Comments 2: To enhance the clarity and utility of your review, I suggest including a table that outlines the main parameters for defining arrhythmic risk in Anderson-Fabry disease, ordered by their importance. It would be helpful to cite relevant references for each parameter in the table.
Response 2: Agree. We have, accordingly, included a table that outlines the main parameters for defining arrhythmic risk in Anderson-Fabry disease with references.
Comments 3: If you wish to continue providing a more general narrative review on cardiovascular involvement in Fabry disease, I would recommend incorporating the following recent studies to enhance the comprehensiveness of your review further: ..
Response 3: Thank you for your suggestion. We have incorporated the recommended references into the appropriate sections to enrich the manuscript and provide a more comprehensive perspective. We value your insightful feedback.
Reviewer 2 Report
Comments and Suggestions for Authors
I had the pleasure of reviewing Tognola et al.'s abstract about arrhythmia and Fabry Disease. The paper focuses on Anderson-Fabry disease (AFD), a rare disorder that causes arrhythmias and other organ issues. It reviews current methods for assessing arrhythmic risk, including diagnostic tools like echocardiography and ECG monitoring, as well as genetic markers. It also discusses treatments such as enzyme replacement therapy and antiarrhythmic drugs, while highlighting the importance of a multidisciplinary approach to improve patient care and outcomes. This is a very good manuscript, and I have some small comments and suggestions.
Major comments:
- I would suggest combining some of the tables about cardiac follow-up for patients. Proband's on enzyme replacement therapy (ERT) usually have a follow-up that is "mandatory" for the drug, which already includes annual (or biannual if <35 years) ECG and echocardiogram. Would the authors agree with this follow-up, or do you think, based on this review, there could be changes to it? What about cardiac MRI?
Smaller comments:
- "Mutation" is used throughout the text; this should be changed to "variant" to align with HGVS nomenclature.
- Gb3 has already been defined before line 65, so the parentheses can be removed.
- "Gene" should be italicized in line 175.
- Make sure it is clear in line 157 that these are pathogenic/likely pathogenic variants that diagnose a patient, and not just any variant.
- Line 162 lacks a reference.
- Line 158 refers to "Diagnosis" on Gene Reviews (https://www.ncbi.nlm.nih.gov/books/NBK1292/) for the correct diagnostic methodology, which might differ in males and females.
- Lines 323-325 seem like a very nice summary. Could this be moved earlier in the text?
- The conclusion is in a different font size.
Author Response
Thank you very much for taking the time to review this manuscript. Please find the detailed responses below and the corresponding revisions in track changes in the re-submitted file.
Comments 1: I would suggest combining some of the tables about cardiac follow-up for patients. Proband's on enzyme replacement therapy (ERT) usually have a follow-up that is "mandatory" for the drug, which already includes annual (or biannual if <35 years) ECG and echocardiogram. Would the authors agree with this follow-up, or do you think, based on this review, there could be changes to it? What about cardiac MRI?
Response 1: Thank you for your suggestion. We have, accordingly, included a table showing recommended cardiac follow-up in patients undergoing ERT. Based on our review, we emphasize the role of cardiac MRI as an essential tool for early detection and monitoring disease progression. We have included this in the revised manuscript. Thank you for your valuable feedback.
Comments 2: "Mutation" is used throughout the text; this should be changed to "variant" to align with HGVS nomenclature.
Response 2: We agree that using the term "variant" instead of "mutation" aligns better with HGVS nomenclature. We have, accordingly, replaced "mutation" with "variant" throughout the text to ensure consistency with current genetic terminology. Thank you for pointing this out.
Comments 3: Gb3 has already been defined before line 65, so the parentheses can be removed.
Response 3: We have removed the parentheses around 'Gb3,' as it has already been defined prior to this line.
Comments 4: "Gene" should be italicized in line 175.
Response 4: We have italicized “Gene” in that line as suggested.
Comment 5: Make sure it is clear in line 157 that these are pathogenic/likely pathogenic variants that diagnose a patient, and not just any variant.
Response 5: Thank you for your comment. We have clarified that the variants being discussed are pathogenic or likely pathogenic variants that are used to diagnose a patient, rather than any variant. This distinction has been emphasized to ensure the accuracy of the information.
Comment 6: Line 162 lacks a reference.
Response 6: Thank you for pointing this out. We have added the appropriate reference.
Comment 7: Line 158 refers to "Diagnosis" on Gene Reviews (https://www.ncbi.nlm.nih.gov/books/NBK1292/) for the correct diagnostic methodology, which might differ in males and females.
Response 7: Thank you for your comment. We have revised this section with the reference recommended for the correct diagnostic methodology, highlighting the difference between males and females.
Comment 8: Lines 323-325 seem like a very nice summary. Could this be moved earlier in the text?
Response 8: Thank you for your feedback. We have moved this section earlier in the text to enhance the flow and provide clarity at an earlier stage.
Comment 9: The conclusion is in a different font size.
Response 9: Thank you for pointing this out. We have corrected the font size issue in the conclusion to ensure consistency with the rest of the manuscript.
Round 2
Reviewer 2 Report
Comments and Suggestions for Authors
I agree with the changes. Thank you,